# Indoor Air Quality in Passivhaus Dwellings: A Literature Review

**DOI:** 10.3390/ijerph17134749

**Published:** 2020-07-01

**Authors:** Alejandro Moreno-Rangel, Tim Sharpe, Gráinne McGill, Filbert Musau

**Affiliations:** 1Lancaster Institute the Contemporary Arts, Lancaster University, Bailrigg, Lancaster LA1 4YW, UK; 2Department of Architecture, University of Strathclyde, 75 Montrose Street, Glasgow G1 1XJ, UK; tim.sharpe@strath.ac.uk (T.S.); grainne.mcgill@strath.ac.uk (G.M.); 3Mackintosh Environmental Architecture Research Unit, The Glasgow School of Art, 167 Renfrew Street, Glasgow G3 6RQ, UK; f.musau@gsa.ac.uk

**Keywords:** indoor air quality (IAQ), Passivhaus, indoor environment, thermal comfort, healthy homes, literature review

## Abstract

Indoor air quality (IAQ) is a critical consideration in airtight buildings that depend on mechanical ventilation, such as those constructed to the Passivhaus standard. While previous reviews of IAQ on Passivhaus-certified buildings foccused on offices, this study examines residential buildings. A summary of data collection methods and pollutant concentrations is presented, followed by a critical discussion of the impact of Passivhaus design strategies on IAQ. This review indicates that IAQ in Passivhaus-certified dwellings is generally better than in conventional homes, but both occupant behaviour and pollution from outdoor sources play a significant role in indoor concentrations. Moreover, there are differences in data collection and reporting methods. Many of the available studies depend on short-term IAQ monitoring of less than two weeks, making it difficult to determine the longer impact of housing design on IAQ and occupants’ well-being. There is also a lack of studies from non-European countries. Future research should focus on investigating associations between IAQ and Passivhaus design strategies in hot and humid climates, where evidence is particularly lacking. Further effort is also required to investigate potential links between occupant’s perception of IAQ and physical exposure to indoor pollution. Finally, the lack of homogeneous monitoring and reporting methods for IAQ studies needs to be addressed.

## 1. Introduction

In recent decades, it has become clear that humans are polluting the Earth to a point beyond which natural systems can function, resulting in progressive climate change [1,2]. Sustainable buildings are an important step to reduce these impacts [3]. It is estimated that the built environment is responsible for 40% of global annual final energy [4,5], and the residential sector may be accountable for a significant part [6,7,8,9]. Building practices, such as those adopted by the Passivhaus standard, are evolving and achieving ultra-low-energy consumption and high levels of occupant comfort, whilst producing buildings that are also economical, resource efficient and resilient to climate change.

A Passive House, or ‘Passivhaus’, which is the original German term, is: “[…] a building, for which thermal comfort (ISO 7730) can be achieved solely by post-heating or post-cooling of the fresh air mass, which is required to achieve sufficient indoor air quality conditions – without the need for additional recirculation of air [10].” The Passivhaus standard is based on five fundamental concepts: super-insulation, thermal bridge-free construction, an airtight building envelope, use of high-performance doors and windows and heat recovery ventilation systems. Further, the building must comply with strict design criteria listed in detail on the Passive House Planning Package (PHPP, currently version 9) [11]. In cooler climates, the most crucial factors are heating load and heating demand so that the building does not require conventional heating systems to maintain comfortable indoor environment levels [12]. 

Ventilation in Passivhaus homes, in most cases, is achieved through a balanced system of extracting and supplying fresh air, aligned with heat recovery. Mechanical ventilation with heat recovery (MVHR) is “[…] dimensioned for airflow rates according to IAQ requirements. Also, for IAQ reasons, air recirculation is not considered ([13], p. 1194)”. MVHR systems, when installed, commissioned and operated correctly, can provide acceptable ventilation levels, high levels of comfort and energy reduction while achieving acceptable IAQ [13]. MVHR installation has been associated with lower CO_2_ concentrations [14], improved IAQ [15,16] and thermal comfort [17], as well as energy savings [16,18], especially in Passivhaus residential buildings [19]. However, these outcomes depend on favourable ambient conditions and operating parameters [20].

Between 1990 and 2005, few Passivhaus homes were built, but this number has increased more rapidly in recent years. According to the Passivhaus Trust, the Passivhaus standard is one of the fastest-growing building energy performance systems in the world and it is estimated that there are now over 65,000 Passivhaus buildings worldwide [13]. Over the last decade, interest in the Passivhaus standard has increased along with research to support the approach and ethos. However, most studies focus on engineering, energy and environmental aspects, as the Passivhaus’ main goal is to reduce energy consumption. Relatively little has been done to investigate the interaction between energy efficiency and indoor environmental quality in Passivhaus buildings [21], particularly IAQ in dwellings. Components of IAQ include a number of variables including temperature and moisture, but more specifically pollutants including particulates, volatile organic compounds (VOCs), organic matter such as mould, and other chemicals. Their presence may be affected by sources (e.g., building materials or cooking), but the key mitigating strategy is ventilation. Studies have demonstrated that several parameters influence CO_2_ [22] and investigations into IAQ commonly use CO_2_ as an indicator of ventilation, especially when investigating living environments [21]. 

As the time we spend indoors increases [23], health problems related to indoor air quality have become more evident [24]. Passivhaus adheres to strict levels of airtightness and then relies on use of mechanical ventilation (MVHR) to control ventilation [11], which may impact on IAQ. However, the Passivhaus standard does not explicitly address occupant health, including off-gassing (release of airborne particulates/chemicals) of building materials, as its approach is based on energy consumption and thermal comfort. Therefore, a comprehensive method to assess IAQ requires identification of specific pollutants [25], such as individual volatile organic compounds (VOCs), total volatile organic compounds (tVOCs) and fine particles (PM_2.5_), and uses CO_2_ as a metric for ventilation [26].

Evaluation of the impact of design strategies in low-energy buildings on occupants’ health and on the indoor environment has identified some concerns [27]. This has led to investigations of the impact of controlled ventilation rates [22,28], MVHR systems [29], airtightness [27]—such as those found in Passivhaus dwellings—and high levels of indoor air pollutants [30,31,32] on human health. Considering that Passivhaus offices and schools have the potential to improve energy conservation and IAQ [21], one might expect similar effects in dwellings. People spend more time in homes than in offices or schools [33]. However, IAQ residential guidelines and policies are not well developed and most of the criteria used for IAQ assessments are based on studies about the effects of air pollutants in non-residential buildings. Many homes contain indoor air pollutant sources such as cooking, cleaning products, tobacco smoke, air fresheners [23,32] that differ from non-residential buildings. Indoor VOCs and PM_2.5_ in homes are often found in higher concentrations than in offices and schools [26,34,35]. As well as the buildings themselves the way that homes are occupied and used, will also affect IAQ performance. 

Over the last decade, reviews of associations between design approaches to Passivhaus buildings and IAQ has been sparse, primarily based on non-residential Passivhaus buildings [21] or isolated aspects of the Passivhaus design strategies based only on occupant’s perceptions [36]. For instance, a review published in 2015 [36] did not find any evidence to suggest that the use of air-heating in Passivhaus homes was negatively affected by levels of IAQ and thermal comfort, but did note some limitations in terms of thermal comfort, especially in bathrooms and bedrooms and general complaints of dry air during winter. However, drawing conclusions from studies based on occupants’ perception of the effectiveness of air-heating, they report that air-heating in residential Passivhaus buildings is possible without adverse effects in health or comfort. 

Passivhaus’ rigorous design and construction methods, along with post-completion testing and verification, especially those related to the building fabric (i.e., airtightness testing), are key components in ensuring that energy targets are achieved. Therefore, the strict controls used in the construction phase are, in a way, a form of warranty that the building will perform as designed and that its results can be replicated. Rigorous monitoring of quality control during construction and the commitment of the design team are key factors in achieving Passivhaus standards [37]. Passivhaus technologies, including heat and cooling recovery ventilation, passive cooling and pre-heated/cooled fresh air, are very promising and have the potential to not only enhance IAQ, but also improve energy efficiency at the same time [21]. However, more recent studies have examined the relationships between IAQ and Passivhaus with more mixed results: while some suggest that Passivhaus design strategies may be beneficial to the indoor environment [12,38], others have found overheating and high levels of CO_2_ [39]. Some research has been carried out on IAQ in Passivhaus buildings, but there is no single study that currently exists, to the knowledge of the authors, that reviews evidence between 2000 and 2020 from physical IAQ measurements in a residential context and contextualises them with regards to the main Passivhaus design strategies.

This paper provides a current review of the literature on IAQ in Passivhaus-certified dwellings. In doing so, this study aims to evaluate the potential for Passivhaus to provide good IAQ. Section 2 defines the methods and criteria for the literature review of this work. Section 3 reviews the findings and research design of the studies that have investigated IAQ in Passivhaus residential buildings. Section 4 discusses the main Passivhaus design strategies that impact on IAQ. Finally, Section 5 presents conclusions and further work.

## 2. Methods 

Journal publications were identified through searches using Scopus and Google Scholar using different combinations of the following search terms: home, dwelling, indoor air quality, IAQ, Passivhaus, Passive House, tVOCs and PM_2.5_. Papers cited in the peer-reviewed articles were also considered. The goal of the search was to identify literature from studies that measured the concentrations of indoor pollutants and occupant perceptions of IAQ in residential Passivhaus buildings. Excluded from this review were studies based on: (a) non-certified Passivhaus dwellings, (b) non-residential buildings, and (c) studies prior to 2000. Due to the limited amount of published research, exclusion criteria did not include studies that draw conclusions from modelled IAQ, those that only used CO_2_ as an IAQ metric, or studies that drew conclusions exclusively from occupant’s perceptions. Literature reviews from non-residential Passivhaus buildings were scrutinised looking for additional relevant literature for this work, but not referenced in this work.

The location, climate, study aim, data collection method (user surveys, physical measurements and computer simulations), duration and number of dwellings, as well as the type of sensors were compiled and separated based on their findings. The findings of the studies are discussed in this review and contextualised with regards to the main Passivhaus design strategies. The reviewed studies were too diverse for statistical analysis. Consequently, the strengths and limitations of the research design of each study were carefully analysed. Comparative tables and texts are used as the methods of evaluating and synthesising the reviewed articles.

Through the manuscript, IAQ is described as high medium, moderate and low accordingly to the descriptions in the “CIBSE Guide A: Environmental Design“ and the “BS EN 13779: Ventilation for buildings. Performance requirements for ventilation and air-conditioning systems”. See Table 1. The term “acceptable” is understood as ‘air in which there are no known contaminants at harmful concentrations as determined by cognizant authorities and with which a substantial majority (80% or more) of the people exposed do not express dissatisfaction [40] (p. 3)’ in this work.

## 3. Results

### 3.1. Studies of IAQ in Passivhaus Dwellings

After exclusions based on the above criteria, forty studies were identified that provided data about IAQ in Passivhaus dwellings. The studies are listed in Table 2 and Table 3. 

### 3.2. Main Findings 

Twenty-four of the forty studies concluded that Passivhaus dwellings have the means to achieve acceptable levels of IAQ [13,41,42,44,45,49,53,54,56,58,61,64,69,70,71,72,75,77]. Eleven studies that compared Passivhaus to conventionally built dwellings found better levels of IAQ in the Passivhaus alternatives [41,45,52,54,58,59,66,70,71,72,77]. The Passivhaus standard does not specifically address off-gassing (release of airborne particulates/chemicals) of building materials, as its approach is based on energy consumption and thermal comfort. However, overheating problems and dry indoor environments have also been reported [46,49,64,65,66].

Twenty-seven of the fourty IAQ studies in Passivhaus dwellings have been undertaken in cold, oceanic, maritime and Mediterranean weathers, mostly in European countries. Very few were carried out on warm and humid climates. Only seven of the studies were carried out in non-European countries [41,42,56,67,69,73]. This demonstrates the need to address IAQ studies worldwide focusing on different climates. 

One of the biggest challenges to compare IAQ between different studies is the lack of homogeneous methods to report IAQ. Some studies do not describe the energy performance (four studies: [55,56,72,77]) or describe IAQ absolute values (fifteen studies: [43,44,46,50,51,57,58,61,65,66,67,68,69,70,75]) and express relative levels (i.e., percentage of time above or below particular thresholds)—which also happen to vary from one study to another, in addition to the differences of monitored periods. Additionally, the timeframe on which studies are conducted varies. For instance, some studies were carried out over long periods in several houses, but measurements were only taken for a week or spot measurements and were non-simultaneous. Many studies discuss results based on short monitoring periods below two weeks ([42,45,48,53,54,55,56,59,60,61]) and others do not consider seasonal variation ([41,42,47,56,61,63]).

The identified literature includes four studies that reported from virtual simulations [47,51,52,66], one study based on occupant perception of IAQ, three used low-cost monitors [41,44,56], twenty-nine included physical IAQ measurements [41,42,43,44,45,48,50,53,54,55,56,57,58,59,60,61,63,64,65,66,68,69,70,71,72,73,74,75,76,77] and fourteen studies assessed IAQ through both physical measurements and occupant perceptions [41,42,43,55,56,58,59,64,65,66,72,73,76]. Only thirteen of these measured IAQ metrics other than CO_2_ [41,45,53,54,55,56,59,66,71,72,73]—of which, only seven also investigated the occupant IAQ perception [41,55,56,59,66,72,73]. It was noted that studies that monitored IAQ parameters other than CO_2_ only collected data between one spot measurement and two weeks of on-site analysis, with the exception of one study [41]. Table 4 shows a summary of the main findings and suggests actions or further work.

### 3.3. IAQ Performance in Passivhaus Dwellings

Passivhaus building systems not only help to achieve low-energy consumption, but they should also provide favourable IAQ and healthier environments. To achieve these aims, it is critical to adhere to best practices in terms of design through to construction and even occupant education [78,79]. Other simulations and field research indicate that Passivhaus design strategies may have a detrimental impact on IAQ [80,81,82]. Perhaps the most significant challenge for energy-efficient buildings related to IAQ is the lack of conclusive evidence.

The quality of the indoor environment in newly built Passivhaus dwellings is comparable or better than other new low-energy homes, especially in relation to IAQ, as buildings achieved higher air change rates [54]. Concentrations of total volatile organic compounds (tVOCs), particulate matter (PM_2.5_) and formaldehyde were found to be lower in Passivhaus dwellings, but dryer environments were also observed [41,54]. Dryer environments have also been reported in other Passivhaus dwellings, especially during winter [66,71,72], associated with high temperatures [66] and the use of MVHR systems [72]. However, simulation and laboratory studies demonstrated that pre-heated air had no adverse effects on IAQ or thermal comfort and was associated with high occupant satisfaction [36]; therefore, occupant behaviour and incorrect use of the system may lead to dry environments.

Passivhaus dwellings should achieve acceptable IAQ by following the mandatory certification criteria. These can be easily enhanced by including the best IAQ practices for source control, local exhausts, continuous ventilation, filtration, commissioning and occupant education [56]. However, overall IAQ performance is also affected by outdoor air quality, indoor emissions, ventilation use and maintenance, and air exchange rates. Human activities have been found to increase alkanes, benzene, aldehydes and PM_2.5_ temporarily, compared to the pre-occupancy period in Passivhaus dwellings, but mean indoor pollution emissions from building materials are generally higher during pre-occupancy and decrease over time [45]. There is some concern about the effects of human activities and behaviours in Passivhaus homes. Measured indoor pollutants in pre-occupied Passivhaus dwellings are usually low; therefore, Passivhaus dwellings with air change rates of 0.5 h^−1^ have the potential to achieve good IAQ [48]. In fact, “the variance of almost all… indoor air pollutants can be explained by their outdoor concentrations and the presence of human occupants and their related activities rather than by building characteristics ([54], p. 90).” However, geographical location and building characteristics may have an impact on indoor temperature, relative humidity, air exchange rate and concentrations of formaldehyde [55]. 

Research on radon in Passivhaus dwellings has gained interest in 2019 [74,75,77]. Although radon gas concentrations in recently renovated dwellings is significantly higher compared to older buildings, there is no difference between non-Passivhaus and Passivhaus dwellings [77]. However, new built Passivhaus dwellings have reduced radon gas concentrations [74] due to the ventilation systems [75]. All of the studies produced equivalent results, with Passivhaus dwellings achieving lower primary energy consumption (42–90% lower) and CO_2_ emissions (25–78% lower) when compared to conventional buildings, with IAQ being observed as acceptable. 

## 4. IAQ and Passivhaus Design Strategies 

### 4.1. Airtightness

The use of airtightness in Passivhaus buildings serves two primary purposes: energy conservation and protection of the building fabric [12]. Leaking building envelopes may lead to a series of problems, such as water damage by condensation, draughts, cold air above the floor level and increased energy consumption. High levels of airtightness, such as those in Passivhaus structures (≤0.6 h^−1^ @50 Pa), may help to avoid condensation and conserve energy. However, studies have opposing results as to whether air infiltration may be either beneficial [36,56,83] or detrimental [27,84,85,86,87,88] for buildings occupants’ health.

A study [56] that measured IAQ and several indoor air pollutants in 19 homes in California found that IAQ was better in those that had higher levels of airtightness. The Passivhaus dwellings were the tightest, but they also had the best practices to control IAQ. However, they noted that if these practices—source control, local exhaust, continuous ventilation, filtration, commissioning and occupant education—were not included, IAQ may be compromised to some extent. Another study [89] that looked at two homes with n_50_ of 0.89–1.60 h^−1^ and mechanical ventilation, and a control house with n_50_ 7.13 h^−1^ and natural ventilation, found no differences in the concentrations or composition of PM_2.5_. Another study suggests that when poor airtightness allows air to be drawn in from contaminated areas, IAQ can be reduced, as the infiltrating air is unfiltered, and in some cases, the building envelope may be a source of pollution because of mould or toxic materials [83].

As energy-efficient homes are made more airtight, indoor pollution sources may be more prevalent. Therefore, adequate ventilation can be used as a strategy to control air pollution [90] becoming critical to achieve and maintain satisfactory levels of IAQ [27], as less reliance can be placed on the building’s air permeability to contribute to air changes [91]. The provision of ventilation is therefore imperative, as there are consequences for the health of occupants when adequate ventilation is not achieved [22,92,93]

### 4.2. Ventilation Rates

Removing indoor sources of air pollution is a key strategy for maintaining good IAQ [93]. In Passivhaus dwellings, MVHR systems are used mainly for provision of fresh air to occupants, but they also act as a way to contain, dilute and remove indoor pollution and moisture [28,94]. For instance, the quality of the air in Passivhaus dwellings was compared with other low-energy homes and conventional houses in Sweden [54]. The study found that while tVOCs were slightly higher in Passivhaus dwellings (but not significantly different from other houses), concentrations of specific VOCs and formaldehydes were lower. Passivhaus dwellings were also characterised by the absence of microflora related to mould, thereby indicating a comfortable and healthier indoor environment. The study suggests that the better IAQ in Passivhaus residences is down to their relatively high air exchange rates. 

Reducing ventilation rates is likely to affect human health [95]. As explained by Wargocki ([23], p. 111): “Ventilation rates above 0.4 h^−1^ or CO_2_ below 900ppm in homes seem to protect against health risks[…], as ventilation rate in homes is associated with health in particular with asthma, allergy, airway obstruction and SBS symptoms[…]. Increasing ventilation rates in homes reduce house dust mites known to cause allergic symptoms.” The commonly accepted threshold [96] below which associations may occur is 0.5 ach^−1^, which should help to control moisture, but may differ from other widely known thresholds (CO_2_ < 1000 ppm or 8 l/s) [97]. Passivhaus ventilation rates are set according to the German standard DIN1946-6 [98], which establishes flow rates between 0.5 and 1.0 ach^−1^. The mean ventilation rates for Passivhaus structures are determined for IAQ requirements, with the minimum being a supply flow of 30 m^3^/h (8.33 l/s) per person, thus allowing the system to have at least 0.2 h^−1^ air changes when there is no occupancy in the building [13]. Evidence shows that ventilation rates in homes below 0.5 ach^−1^ may degrade occupants’ health, as they are associated with a higher likelihood of exacerbating the symptoms of asthma and allergies from indoor pollutants [26]. 

Data are limited regarding the causal health effects associated with ventilation rates in houses [99]; however, “as the limit values of all pollutants are not known, the exact determination of required ventilation rates based on pollutant concentrations and associated risks is seldom possible [28]”. Different studies suggest that low ventilation rates not only result in increased concentrations of indoor-generated pollutants, but they are also associated with SBS symptoms, poor thermal comfort, negative health effects and reduced productivity in non-industrial buildings [87,100]. An increase in SBS symptoms was associated with low ventilation, with human responses to low ventilation rates likely to affect IAQ perceptions and productivity [28], causing inflammation, asthma, allergies and short-term sick leave in office buildings [100].

There is a wide range of research findings on whether Passivhaus ventilation rates might be appropriate to maintain acceptable IAQ. For instance, it has been reported that Passivhaus with air change rates of 0.5 h^−1^ has the potential to achieve good IAQ [48]. Others suggest that while Passivhaus ventilation may be sufficient to comply with regulations or provide occupants with breathable air, it might not be enough to remove concentrations of VOCs, particulates and other hazardous chemicals [101]. Low ventilation rates [92] and dampness [102] have been associated with asthma, rhinitis and eczema in Swedish homes, so higher ventilation rates are highly desirable.

A comparison between the USA, European and Passivhaus ventilation standards found an apparent lack of ventilation guidelines for Passivhaus [52]. Ventilation rates (8.3–8.9 l/s per person) required for Passivhaus dwellings account for the entire building only, whereas local guidelines might suggest different air flows (exhaust and supply), depending on the room. However, perhaps this opens up the possibility for Passivhaus to adapt to local regulations.

A frequent practice is to use CO_2_ as an indicator of ventilation rates [22,26], and levels below 1000 ppm are associated with adequate solutions in this regard [97]. Passivhaus studies that have measured CO_2_ concentration often report a wide range of values. For instance, measured CO_2_ concentrations in Romanian Passivhaus homes were below 800 ppm [44] and below 1000 ppm (between 810 and 832 ppm) in US Passivhaus dwellings [42]; exceptions were when the house was occupied with more people than for what it was designed (hosting a dinner party, for instance). Another study [65] measured CO_2_ concentrations in two Passivhaus homes in Wales for over two years. The dwellings were designed to meet the EN 13779 [103] “moderate or satisfactory” IDA3 category (CO_2_ levels above the range 600–1000 ppm outdoor air, <1400 ppm). According to IDA3, houses should have ventilation greater than 3.33 l/s per person. In one of the houses, the MVHR unit met 6.93 l/s per person, and bedroom CO_2_ concentrations exceeded 1400 ppm over 12.9% and 1000 ppm over 36% of the time over the two years. The second dwelling achieved a ventilation rate of 11.31 l/s per person, and bedroom CO_2_ levels exceeded 1400 ppm only 0.1% and 1000 ppm over 9.5% of the time over the two years. Eight Passivhaus flats were compared to eight conventional flats in China [73]. The authors found that ventilation levels of 8.33 l/s per person or higher were sufficient, thereby concluding that Passivhaus dwellings achieve acceptable CO_2_ levels. CO_2_ concentrations in the Passivhaus flats were between 622 and 841 ppm, whereas four of the conventional flats in the study exceeded 1000 ppm.

Other studies present contradictory evidence. For instance, a study that measured three Passivhaus units in Denmark found that winter CO_2_ levels were above the target (660 ppm above the outdoor (outdoor average 370 ppm)), while summer CO_2_ levels were acceptable. During winter, CO_2_ thresholds were exceeded in two of the homes [43]. However, the authors noted that occupants normally opened their windows during summer. UK Passivhaus dwellings may have poor ventilation, especially social housing [60]—although Passivhaus standard is not a common practice for UK social housing—as the CO_2_ thresholds were often exceeded when the rooms were occupied. However, they concluded that this could be down to some deficiencies in the MVHR system, including a lack of occupant knowledge.

### 4.3. MVHR Systems

An MVHR is a ‘whole-house’ ventilation system, in which fresh air circulates from the supply zones to the extract zones so that the whole house is continually refreshed with clean, filtered outdoor air. The heat recovery element is the key factor for this ventilation strategy, as the incoming air is pre-heated by the extracted air on a counter-flow heat exchanger chamber without mixing them. There are several components of the MVHR systems, but perhaps the most important for IAQ are the ducts, supply and extract terminals, as well as the filters. Although the main purpose of filters is to protect the heat exchange unit from dust, they may provide some protection from solid air pollutants as a secondary effect. However, the correct filters must be used to protect the system components and reduce indoor exposure to pollutants from outdoor origin. For instance, Passivhaus employing grade G4 filters and without secondary filters instead of the F7, as required for the certification, inadequately filtrated outdoor PM_2.5_, resulting in higher indoor concentrations [89].

Limited data are available on whether the effectiveness of MVHR systems to provide ventilation and control IAQ is adequate or not. Some studies suggest that they may actually exacerbate, rather than resolve, IAQ problems [101]. A significant concern of sizing residential MVHR units has been noted in current Passivhaus practices, as they deliver the same background ventilation regardless of occupancy levels [50]. It is clear that in order to benefit from the above, MVHR systems should be adequately designed, commissioned, installed, maintained and operated. A recent study of 54 homes in the UK, in which MVHR systems often did not perform as intended, found numerous problems related to installation, commissioning stages, operation and performance [104]. These findings are similar to earlier studies investigating MVHR deficiencies [105,106,107]. McGill [59] suggest that most of these problems could be avoided at the design stage. If proper instructions and guidance are given, problems in installation and commissioning could be prevented, thus averting problems with operation and performance. Recent studies have also found incidents of overheating in Passivhaus [104,108,109], complaints regarding the noise of the MVHR [62,91,104,105], cold draughts [104] and occupants’ experiences when interacting with the ventilation unit [70]. These problems may lead to the intermittent or seasonal use of MVHR systems as one of the many occupant responses to such deficiencies. MVHR performance shortcomings in Passivhaus projects were observed less often than in homes without the certification [104], due to the rigorous certification process. However, despite the shortcomings listed above, MVHR systems could result in higher levels of ventilation and lower energy consumption compared to naturally ventilated houses, but the context for this may be even worse ventilation in non-MVHR houses [104].

MVHR systems, regardless of the building type in which they are installed, are more energy efficient, with higher levels of airtightness [110,111]. However, this raises other issues, as mechanical ventilation systems have been associated with VOCs and other chemical pollutants emitted by system components and ductworks [28]. 

Naturally ventilated and MVHR-equipped dwellings were studied to find associations between SBS symptoms, CO_2_ and formaldehyde levels [72]. They found that associations between neurological symptoms (dizziness, nausea and headaches) and formaldehyde concentrations as well as between CO_2_ levels and perceived stale air were observed. However, both associations were observed regardless of the type of ventilation. Recent studies, however, have demonstrated the difficulties involved in regular maintenance and cleaning—for instance, the limited options for filter replacements for ventilation units in the UK [32]. A study [106] that looked at 150 homes with MVHR systems found that the most common problem was general maintenance and cleaning. In total, 66% of the homes did not undertake annual maintenance, visible dirt was found in 43% of the homes, 77% had dust and dirt on the ducts and 67% had visible dirt from material construction. Occupant interaction with the system is a critical dimension. Inadequate user understanding and awareness of MVHR operation and control [70], combined with habitual behaviours (i.e., unexpected window openings), leads to misuse [112].

These studies have described the possible implications of the Passivhaus design strategies for IAQ and occupants’ well-being. However, airtightness, ventilation rates and MVHR systems should be understood as one entity in Passivhaus dwellings in order to provide a deeper understanding of the level of protection achieved following the rigorous criteria for certification. 

## 5. Conclusions

Passivhaus design strategies (airtightness, controlled ventilation rates and MVHR systems) have the potential to achieve substantial energy reductions and good levels of IAQ, but only if building professionals and occupants seek to adhere to the best IAQ practices. As a function of its additional complexity and reliance on mechanical systems, occupants of Passivhaus dwellings need a greater degree of awareness and education to ensure the quality of their indoor environments.

One of the biggest challenges when comparing IAQ studies, such as this, is the differences in monitoring periods, ways of reporting the data, the variety on indoor air pollutants measured/IAQ metrics, and the lack of universal thresholds. The latter, in particular, made it difficult to compare studies that only reported the percentage of time exceeding acceptable thresholds. Another characteristic is that the available IAQ studies often consider the indoor environment over a very limited time frame (e.g., one spot measurement to a maximum of two weeks). Despite the recent evidence on the impact of IAQ on health, very few Passivhaus studies link occupants’ well-being and IAQ perceptions to physical concentrations of indoor air pollution.

This review indicates there are gaps in the knowledge. There is a need to standardise IAQ assessment—frequency and range of pollutant types—recognising that CO_2_ is a proxy for ventilation rather than an IAQ indicator. The best practice observed in this work entailed obtaining high-quality data simultaneously in different buildings over a more extended time frame which could be used as a starting point to a more uniform IAQ reporting procedure. This reporting procedure could make comparative studies across multiple build types and climates, and facilitate research to observe potential links between occupants’ IAQ perceptions and well-being, to physical exposure of indoor pollution levels and indoor environmental parameters. These outcomes highlight a lack of studies addressing IAQ in Passivhaus homes and the need to achieve a better understanding of the impact of Passivhaus design techniques on IAQ.

## Figures and Tables

**Table 1 ijerph-17-04749-t001:** Definition of indoor air quality (IAQ) standards.

Category	IAQ Standard	Ventilation Range (l/s per Person)	Default Value (l/s per Person)	CO_2_ Level above the Outdoor	
				Typical Range (ppm)	Default Value (ppm)
IDA1	High	>15	20	≤400	350
IDA2	Medium	10–15	12.5	400–600	500
IDA3	Moderate	6–10	8	600–1000	800
IDA4	Low	<6	5	>1000	1200

**Table 2 ijerph-17-04749-t002:** Publications about IAQ in Passivhaus dwellings. Part A.

	Geographic Location										Data Collection Method													No. Homes		
	Location	Climate									Surveys				Physical Measurements											Total
	Country	Cold	Temperate	Continental	Oceanic	Warm/humid	Mediterranean	Subtropical	Maritime	Humid	Well-being	Thermal comfort	IAQ	Other	Temperature	Relative humidity	Absolute humidity	CO_2_	PM_2.5_	tVOCs	Specific VOCs	Other	Outdoor data	Passivhaus	Other	
[12]	Several A **	●	●	●	●	●						●	●		●							●		100		100
[13]	Several A **	●	●	●	●										●	●						●		100		100
[38]	Several A **	●	●	●	●	●						●	●		●							●		100		100
[41]	Mexico					●					●	●	●		□	□		□	□				●	1	1	2
[42]	USA											●	●	●	●	●		●				●	●	6		6
[43]	Denmark		●									●	●	●	●	●		●				●		3		3
[44]	Romania		●												□	□		□				●		1		1
[45]	France			●	●		●								●	●		●	●	●	●	●		1	6	7
[46]	Portugal						●								●								●	1		1
[47]	Poland		●												◊							◊		1		1
[48]	Sweden				●															●	●	●	●	1		1
[49]	Cyprus							●							●	●								1		1
[50]	Scotland		●		●									●	●	●	●	●				●	●	2		5
[51]	Norway	●													◊							◊		1		1
[52]	Several B **		●	●	●		●		●													◊		1	5	6
[53]	Lithuania			●	●									●	●	●		●		●	●	●	●	●	●	11 *
[54]	Sweden				●										●	●		●		●		●		20	21	41
[55]	France		●	●	●								●	●	●	●		●	●	●	●	●	●	●	●	567 *
[56]	USA						●						●	●	●	●		●	□			●	●	●	●	24 *
[57]	England		●						●						●◊	●		●					●◊	1	1	2
[58]	Austria				●						●	●	●		●	●		●						2	2	4
[59]	England		●						●		●	●	●	●	●	●		●			●		●	2	5	7
[60]	England		●						●				●	●	●	●		●					●	3		3
[61]	Romania			●						●					●◊	●		●				●◊		1		1
[62]	Netherlands								●			●	●	●										7	83	90
[63]	Denmark		●											●	●◊	●◊		●◊				●◊		1	1	2
[64]	England		●						●			●	●	●	●	●		●				●	●	1		1
[65]	Wales								●			●	●	●	●	●		●				●	●	2		2
[66]	Austria			●						●		●	●		●	●		●		●			●	18	6	24
[67]	Several C **	●				●		●							◊	◊								7		7
[68]	Scotland		●		●										●	●	●	●				●		5	21	26
[69]	Australia				●										●			●				●		1		1
[70]	Scotland		●		●										●	●		●				●	●	1	2	3
[71]	Austria				●										●	●		●		●	●	●		●	●	123 *
[72]	Austria				●						●	●	●	●	●	●		●		●	●	●		●	●	123 *
[73]	China	●											●	●	●	●		●	●			●		8	8	16
[74]	N. Ireland				●																	●		5		5
[75]	Germany								●					●	●	●		●				●	●	4		4
[76]	Norway	●										●	●	●	●	●		●				●		1		1
[77]	Germany								●													●		114	41	155

* Number of dwellings not described by energy performance, but Passivhaus in homes sampling. ** Several A: Austria, Germany, France, Sweden, and Switzerland. Several B: Germany, France, Spain, and the UK. Several C: Russia, Japan China, the USA, Singapore, and the United Arab Emirates. ● = physical monitoring with analytical monitors or not specified; □ = physical monitoring with low-cost monitors; ◊ = computer simulation; y = year(s); m = month; d = day(s).

**Table 3 ijerph-17-04749-t003:** Publications about IAQ in Passivhaus dwellings. Part B.

					Temporality					Results							
	Room					Season				Surveys *		Physical Measurements					
	Bedroom	Kitchen	Living/open plan	Other	Time span	Winter	Spring	Summer	Autumn	Thermal perception	IAQ perception	Temperature (°C)	Relative humidity (%RH)	Absolute humidity	CO2 (ppm)	PM2.5 (µg/m3)	TVOCs (µg/m3)
[12]	Not described				2.5y	●	●	●	●	V	V	20 (17–27)					
[13]	Not described				2.5y	●	●	●	●	V	V	20 (17–27)					
[38]	Not described				2.5y	●	●	●	●	V	V	20 (17–27)					
[41]	●	●	●		3m			●		G	G	23 (9–29)	52 (35–74)		436 (218–1431)	17.87 (2.5–146.6)	
[42]			●		1w	●				G	V	19 (16–27)	◊		820 (410–2378)		
[43]					3y	●		●		V	G	◊	◊		◊		
[44]			●		2y	●	●	●	●			◊	◊		◊		
[45]	●	●	●		2w	●		●				21 (17–27)	42 (24–59)		887 (331–2030)	16.6	184
[46]	Not described				3m	●		●				◊					
[47]	●	●	●	●	1y	●						20					
[48]					1w												150
[49]	●		●	●	11m	●	●	●	●			24 (16–33)	53				
[50]	●	●	●		1y	●	●	●	●			◊ (18–25)	◊	◊	◊		
[51]	●	●	●	●	1y	●	●	●	●			◊ (19–34)					
[52]					1y												
[53]			●		7d		●	●	●			22	51		673		296
[54]	●				2w	●	●	●	●			22	30		540		272
[55]			●		7d	●	●	●	●			Not described by dwelling’s energy performance					
[56]	●	●			6d				●			Not described by dwelling’s energy performance					
[57]			●		1y	●	●	●	●			22	46		◊		
[58]	●		●		5m	●	●	●		G	G	◊	◊		◊		
[59]	●		●		1d	●		●		G	V	23 (20–25)	41 (26–52)		133 (436–976)		
[60]			●		1d	●		●			G	22 (19–25)	43 (32–53)		731 (396–2598)		
[61]	●	●	●	●	6m	●						◊	◊		◊		
[62]																	
[63]	●	●		●	30d	●						23	35		◊		
[64]	●		●		1y	●	●	●	●			22	49		893		
[65]	●	●	●	●	2y	●	●	●	●			22	◊		◊		
[66]	●		●		2y	●		●		G	G	23	◊		◊		
[67]				●	1y	●	●	●	●			◊	◊				
[68]	●				7m	●	●	●				◊	◊	◊	◊		
[69]	●		●	●	1y	●	●	●	●			◊			◊		
[70]			●		1y		●	●				◊	◊		594 (401–1384)		
[71]					2y	●	●		●			Not described by dwelling’s energy performance					
[72]					2y	●	●		●			Not described by dwelling’s energy performance					
[73]			●		5m	●	●				OK	26 (23–28)	31 (18–46)		732 (622–841)	92 (47–127)	
[74]	●		●		3m	●			●								
[75]					25y	●	●	●	●			◊	◊		850		
[76]			●			●		●		V	V	22 (21–24)	37 (17–61)		383		
[77]					1y	●	●	●	●								

Mean (min–max); ◊ = absolute values not described; G = good; V = very good; OK = neither good or poor. * Perception data are often reported using scales; % of persons dissatisfied was not described in papers. Data refer to Passivhaus homes only; articles were scrutinised to differentiate these data from other types of buildings reported (i.e., control homes).

**Table 4 ijerph-17-04749-t004:** Summary of main findings.

Factor	Practice Observed	Suggestion/Further Work
IAQ monitoring	Lack of homogeneous methods to report IAQ, due to apparent differences in parameters, timeframe, and reporting findings.	Uniformity of IAQ monitoring.
IAQ parameters	More than 50% of the studies that measured IAQ only use CO_2_ as an IAQ indicator.	Further work should include monitoring of specific IAQ pollutants, such as VOCs, PM2.5, CO, formaldehyde.
Timeframe IAQ monitoring	Differences from pollutant measurements vary from one spot measurement, less than 12 hours, a day, a week, a month and a year.	Establish a minimum time frame to measure and report whether the measured time is longer than the minimum report both.
Relation to other monitoring in the same study	More than 90% of the studies did not collect pollution data in homes (excluding CO_2_) simultaneously.	Use of simultaneous measurements.
IAQ reporting	Some studies report absolute values and other relative values. Some studies do not report differences between dwelling types.	Standardise reporting method. Always describe absolute values and, if needed for trends or other analysis, relative values.
Instrumentation	More than 90% used highly accurate monitors. Less than 10% use low-cost solutions.	The use of low-cost monitors could help to overcome the initial costs facilitating simultaneous monitoring, as well as wider timespans and collection samples.
Geographical location	More than 90% of the studies focused on European countries	Conduct IAQ analysis in non-European countries.
Climates	European climates are well represented, but studies in other climates are lacking, such as warm or humid locations.	Conduct IAQ analysis in climates not represented in European locations.

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
