# Peer review of "Indoor Air Quality in Passivhaus Dwellings: A Literature Review"

_ijerph, 2020, doi:10.3390/ijerph17134749_

Round 1
Reviewer 1 Report
The paper by Moreno-Rangel et al “Indoor air quality in Passivhaus dwellings: a literature review” discussed air quality studies for a special type of house -- the Passivhaus (no additional recirculation of air). The authors presented a nice table (table 1) summarizing the details of all available published studies. The authors focused a lot on the indoor CO2 levels, to me, CO2 levels were less important compared to particulate matter PM2.5 and volatile organic compounds (VOC) in the context of air quality. And the current hot topics of indoor air are VOC, particle number, particulate matter, etc. The trend is not quite discussed in this paper, although I understand the author chooses a special type of house. Choices of words are sometimes vague, like acceptable, higher, good, better (e.g., L149). The authors should address this part.
Introduction
What fraction of homes are passive houses? A big picture introduction of passive houses should be included here.
L117 tVOC and PM2.5
Explanations for these acronyms are needed at their first appearance in the paper.
Reviewer 2 Report
Further work identifies the need for more research, and outside of Europe. It would be useful to gather the outcomes in a table, so that the critical points could be captured, this could also indicate the gaps in knowledge, and possibly the impact of occupant behaviour on the running of PH dwellings. Studies over the years on occupant behaviour have an influence here for further work.
A table summarising findings would help as the text in places tends to be descriptive. These are complex issues that could be simplified and contextualised with more discursive text. Although this does not detract from the findings, just doesn't make them as significant as they could be.
There are some key names missing in the references that I would have expected to see, but they may have been omitted as their work tends towards non domestic monitoring - although this is also mentioned in the text, that drawing on and comparison to non domestic PH buildings was used as background to the study. It isn't clear how additional papers were used or identified.
Overall a useful paper to map and the 3 key areas/outcomes for further work are non controversial.
Reviewer 3 Report
A well-organised, grammatically sound and timely piece of work that contains interesting findings and merits publication subject to some additional improvements as outlined below:
Section 3 is quite comprehensive but perhaps can benefit from more organisation. This is a matter of entirely cosmetic preference but I feel it can be titled ‘results’ with perhaps 3 smaller subsections to introduce [3.1] Primary data (where you introduce tables 1A/B [3.2] main findings [3.3] IAQ reports (or any other subdivisions you would like to break the section to. This also relates to my comments on line 156-165).
Well done on maintaining referencing protocol when reproducing exact lines from other literate.
The reporting of ventilation rates in the manuscript contains multiple units, this somewhat is taxing for most readers except those fully proficient in building ventilation science, across lines 275-295 this switching between m3/hr and l/s becomes confusing. I encourage the authors to consider l/s and convert all other units to comply, perhaps with some brief exploratory lines/equations in method section to provide explanation on how all other reported values were converted to l/s.
Specific lines:
13: office…
32: beyond which natural systems ‘CANNOT’ function?
106: Why not say 2003-2020 to cover recent evidence?
136: Base only ‘on’
152: off-gassing: Perhaps worth defining this term or simply opening parenthesis to say (release of airborne particulates/chemicals).
156-165 : These are really interesting and valuable findings. Essentially you are reporting that there is no consistency in the way that – even the scientific community – is reporting its longitudinal observational data on IAQ. This can be a separate sub-section to highlight it more emphatically as outlined in my second paragraph.
227: ‘Adequate’ ventilation is can be used as ‘a’ strategy …
236 in a passivehaus or in passive houses…
279: ‘Passivhaus studies that have measured CO2 concentration often find contradictory evidence’:
What you are reporting (i.e. CO2 concentrations of 800ppm in Romanian /1000pmm in USA homes) arise as a function of the diversity of occupancy patterns and activities. Best to say:
Passivhaus studies that have measured CO2 concentration often report a wide range of values. For instance ….
300-301: UK Passivhaus dwellings may have poor ventilation, especially social housing:
This might surprise some of the readers to learn UK has passivhaus social housing. Perhaps make a brief mention of where these properties are to make your statement more case-specific.
328: …found incidents… or found instances…
360: … but building professionals and occupants must adhere to ….
This sounds very prescriptive and edict-like. You are writing on the basis of the evidence that you have presented and don’t need to try and be persuasive or commanding. Best to say:
‘Passivhaus design strategies (airtightness, controlled ventilation rates and MVHR systems) have the potential to achieve substantial energy reductions and good levels of IAQ, but crucially if building professionals and occupants seek to adhere to the best IAQ practices. As a function of its additional complexity and additional mechanical systems, occupants of passive houses need greater degrees of awareness and education to ensure the quality of their environments.’
These are all findings that the authors have outlined earlier in the draft.
366 … exceeding acceptable thresholds …
371 The authors have identified 3 main areas as a results of their literature review that they begin to report at line 371. This is completely appropriate. However, the first point is slightly nebulous:
‘identifying a need to; (1) compare IAQ between Passivhaus, other low-energy certificates and conventional dwellings, particularly in non-European locations’
I suggest highlighting this with greater clarity and the fact that you have found the instrumentation/monitoring and reporting mechanisms to be very disjointed and non-uniform. And to suggest at least what you have found to be the most appropriate method of reporting IAQ (which you begin to do in point (2) by saying IAQ reporting needs to take place over a long period of time and simultaneously between multiple case-studies representing architypes/climate/etc.
This means that for instance point (1) could be rephrased to say:
‘A great need exists to standardise the duration, frequency and range of pollutant-types that IAQ studies need to report on. The best practice observed in this work entailed … which could begin as a starting point for a more uniform IAQ reporting procedure that makes comparative studied across multiple built-types and climates possible.’
This of course needs to reflect the authors perspective.
Round 2
Reviewer 1 Report
The authors answered my questions.